# BinaryDuo: Reducing gradient mismatch in binary activation network by coupling binary activations

**Hyungjun Kim**[*]**, Kyungsu Kim**[*]**, Jinseok Kim, Jae-Joon Kim**
POSTECH, Department of Creative IT Engineering, Korea
{hyungjun.kim, kyungsu.kim, jinseok.kim, jaejoon}@postech.ac.kr

## Abstract

Binary Neural Networks (BNNs) have been garnering interest thanks to their compute cost reduction and memory savings. However, BNNs suffer from performance degradation mainly due to the gradient mismatch caused by binarizing activations. Previous works tried to address the gradient mismatch problem by reducing the discrepancy between activation functions used at forward pass and its differentiable approximation used at backward pass, which is an indirect measure. In this work, we use the gradient of smoothed loss function to better estimate the gradient mismatch in quantized neural network. Analysis using the gradient mismatch estimator indicates that using higher precision for activation is more effective than modifying the differentiable approximation of activation function. Based on the observation, we propose a new training scheme for binary activation networks called BinaryDuo in which two binary activations are coupled into a ternary activation during training. Experimental results show that BinaryDuo outperforms state-of-the-art BNNs on various benchmarks with the same amount of parameters and computing cost.

## 1 Introduction

Deep neural networks (DNNs) have been achieving remarkable performance improvement in various cognitive tasks. However, the performance improvement generally comes from the increase in the size of the DNN model, which also greatly increases the burden on memory usage and computing cost. While server-level environment with large number of GPUs can easily handle such a large model, deploying a large-size DNN model on resource-constrained platforms such as mobile application is still a challenge.

To address the challenge, several network compression techniques such as quantization (Wu et al., 2018b; Zhou et al., 2016; Lin et al., 2016), pruning (Han et al., 2015; Hu et al., 2016; Wen et al., 2016), and efficient architecture design (Howard et al., 2017; Wu et al., 2018a; Tan & Le, 2019) have been introduced. For quantization, previous works tried to reduce the precision of the weights and/or activations of a DNN model to even 1-bit (Courbariaux et al., 2016; Rastegari et al., 2016). In Binary Neural Networks (BNNs), in which both weights and activations have 1-bit precision, high-precision multiplications and accumulations can be replaced by XNOR and pop-count logics which are much cheaper in terms of hardware cost. In addition, the model parameter size of a BNN is 32x less than its full-precision counterpart.

Although BNNs can exploit various hardware-friendly features, their main drawback is the accuracy loss. An interesting observation is that binarization of activation causes much larger accuracy drop than the binarization of weights (Cai et al., 2017; Mishra et al., 2018). In particular, it can be observed that the performance degradation is much larger when the bit precision is reduced from 2-bit to 1-bit than other multi-bit cases (>2-bit) as shown in Table 1. To explain the poor performance of binary activation networks, the gradient mismatch concept has been introduced (Lin & Talathi, 2016; Cai et al., 2017) and several works proposed to modify the backward pass of activation function to reduce the gradient mismatch problem (Cai et al., 2017; Darabi et al., 2018; Liu et al., 2018).

---

[*]Hyungjun Kim and Kyungsu Kim equally contributed to this work.

Table 1: Classification accuracy of quantized neural network on ImageNet dataset.

| WRPN AlexNet 2x-wide (Mishra et al., 2018) | | | | | ABC-Net ResNet-18 (Lin et al., 2017) | | | |
|---|---|---|---|---|---|---|---|---|
| W\A | 32b | 8b | 4b | 2b | 1b | W\A | 5b | 3b | 1b |
| 32b | 60.5 | 58.9 | 58.6 | 57.5 | **52.0** | 5b | 65.0 | 62.5 | **54.1** |
| 8b | - | 59.0 | 58.8 | 57.1 | **50.8** | 3b | 63.1 | 61.0 | **49.1** |

In this work, we first show that previous approaches to minimize the gradient mismatch have not been very successful in reducing accuracy loss because the target measure which the approaches tried to minimize is not effective. To back up the claim, we propose a new measure for better estimation of gradient mismatch. Then, we propose a new training scheme for binary activation network called *BinaryDuo* based on the observation that ternary (or higher bitwidth) activation suffers much less from gradient mismatch problem than the binary activation. We show that BinaryDuo can achieve state-of-the-art performance on various benchmarks (VGG-7, AlexNet and ResNet). We also provide our reference implementation code online[1].

## 2 RELATED WORKS

**Binary Neural Network** A BNN was first introduced by Courbariaux et al. (2016). In BNNs, both weights and activations are constrained to +1 or -1 and sign function is used to binarize weights and activations. Since sign function is not differentiable at a single point and has zero gradient everywhere else, the concept of straight-through estimator (STE) has been used for back-propagation (Bengio et al., 2013). Courbariaux et al. (2016) used HardTanh function as differentiable approximation of the sign function so that its derivative is used instead of delta function. Rastegari et al. (2016) proposed a similar form of BNN, called XNOR-Net. XNOR-Net used $+\alpha/-\alpha$ instead of +1/-1, where $\alpha$ is a channel-wise full-precision scaling factor.

**Sophisticated STEs** Although using STE enabled the back-propagation of the quantized activation function, gradient mismatch caused by the approximation occurs across the input range of binary activation function. To mitigate the gradient mismatch problem, Darabi et al. (2018) proposed to use SwishSign function as a differentiable approximation of the binary activation function, while Liu et al. (2018) proposed to use the polynomial function.

**Continuous binarization** Few recent work proposed to use a continuous activation function that increasingly resembles a binary activation function during training, thereby eliminating approximation process across activation function. Sakr et al. (2018) used a piecewise linear function of which the slope gradually increases, while Yang et al. (2019) and Gong et al. (2019) proposed to use sigmoid and tanh function, respectively.

**Additional shortcut** Most recent models have shortcut connections across layers. Since the parameter size and computing cost of shortcut connections is negligible, shortcut data often maintains full-precision even in BNN. Bi-Real-Net (Liu et al., 2018) pointed out that these information-rich shortcut paths help improving the performance of BNNs, and hence proposed to increase the number of shortcut connections in ResNet architecture. Since additional shortcuts increase the element-wise additions only, the overhead is negligible while the effect on performance is noticeable.

## 3 SHARP ACCURACY DROP FROM 2-BIT TO 1-BIT ACTIVATION

### 3.1 EXPERIMENTAL CONFIRMATION OF THE POOR PERFORMANCE OF BINARY ACTIVATION

In this section, we first verify that performance drop of a neural network is particularly large when the bit-precision of the activation is reduced from 2-bit to 1-bit (Table 1). We used the VGG-7 (Simonyan & Zisserman, 2014) network on CIFAR-10 dataset for our experiment. We trained the benchmark network with full-precision, 4-bit, 3-bit, 2-bit and 1-bit activation while keeping all weights in full-precision to solely investigate the effect of the activation quantization. Activation

---

[1]https://github.com/Hyungjun-K1m/BinaryDuo

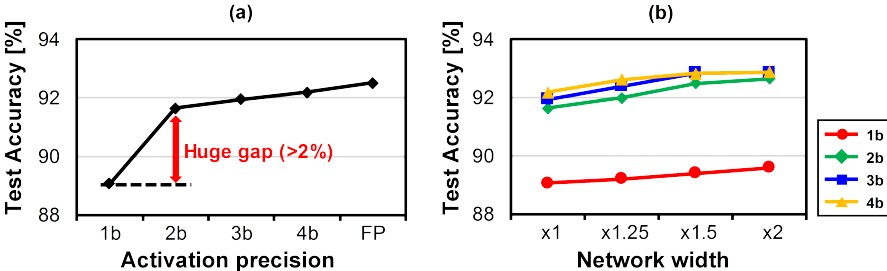

Figure 1: Training results of VGG-7 on CIFAR-10 dataset. (a) Test accuracy with various activation precision. (b) Test accuracy of various activation precision and various network width.

quantization was carried out by clipping inputs outside of [0,1] and then applying uniform quantization as in Eq. 1.

$$Q(x) = \frac{1}{2^k - 1} \cdot \text{round}\big(\text{clip}(x, 0, 1) \cdot (2^k - 1)\big) \tag{1}$$

Here, $x$ and $Q(x)$ are the input and output of the quantization function, and $k$ is number of bits. For consistency, we used Eq. 1 for binary activation function instead of the sign function which is another widely used option (Cai et al., 2017; Zhou et al., 2016; Mishra et al., 2018). We trained each network for 200 epochs and detailed experimental setups are described in Appendix A. The best test accuracy from each precision is plotted in Figure 1a. It is clearly shown that reducing activation bitwidth from 2-bit to 1-bit results in the substantial drop in the test accuracy.

One might argue that the sharp accuracy drop occurred because the capacity of the binary activation network was not large enough for the given task while that of the network with 2-bit or higher bitwidth was. To verify, we increased the model size by modifying the network width (not depth) for each activation precisions (Figure 1b). We found that the large accuracy gap still existed between 2-bit activation network and 1-bit activation network regardless of the network width. For example, 1-bit activation network with 2 times wider layers needs more parameters and more computations, but shows much worse accuracy than 2-bit activation network with original width. In addition, poor performance of binary activation was also observed in different networks (e.g. AlexNet and ResNet) and different datasets as in Table 1. Therefore, we argue that the sharp accuracy drop for the binary activation stems from the inefficient training method, not the capacity of the model.

## 3.2 GRADIENT MISMATCH AND SOPHISTICATED STES

When training a neural network with quantized activation function such as thresholding function, STE is often used for back-propagation. In such case, we call the back-propagated gradient using STE as *coarse gradient* following Yin et al. (2019). However, training with the coarse gradient suffers from gradient mismatch problem caused by the discrepancy between the presumed and the actual activation function (Lin & Talathi, 2016). The gradient mismatch makes the training process noisy thereby resulting in poor performance.

Unfortunately, it is not useful to measure the amount of gradient mismatch directly because the *true gradient*[2] of a quantized activation function is zero almost everywhere. Therefore, previous works used an indirect criterion to measure gradient mismatch based on the discrepancy between the actual activation function and its differentiable approximation (Darabi et al., 2018; Liu et al., 2018). In Liu et al. (2018), the cumulative difference between actual activation function ($f$) and its differentiable approximation ($g$) (Eq.2) was used as a criterion for gradient mismatch.

$$\text{cumulative difference} = \int_{-\infty}^{\infty} |f(x) - g(x)| dx \tag{2}$$

Previous works tried to improve the accuracy by designing the differentiable approximation of binary activation function which has the small cumulative difference from the binary activation function. Figure 2 shows different options for the approximation functions proposed by previous

---

[2]We call the original gradient as 'true gradient' to distinguish with the coarse gradient.

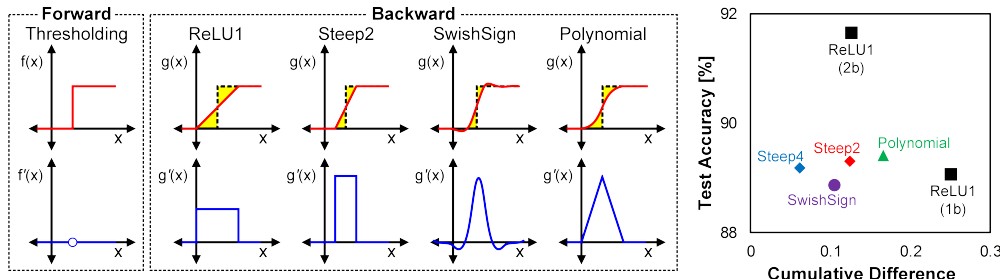

Figure 2: Activation functions and their derivatives used at forward and backward passes (left). The relationship between the best test accuracy and the cumulative difference when using various STEs (right). Steep4 is extended version of steep2 with slope of 4.

works (Darabi et al., 2018; Liu et al., 2018). The shaded area in each activation function represents cumulative difference with thresholding function. We conducted the same experiment for a binary activation network as in previous section with various sophisticated STEs. For each case, the cumulative difference and test accuracy are shown in Figure 2. As proposed in previous works, cumulative differences when using sophisticated STEs are much lower than conventional case (ReLU1). However, its effect on network performance is quite limited in our experiment. As shown in the Figure 2, large accuracy gap still exists between binary activation network and 2-bit activation network even if the sophisticated STEs are used. From this observation, we think that current method of measuring gradient mismatch using the cumulative difference cannot reliably estimate the amount of gradient mismatch problem. The same observation was also reported by Bethge et al. (2019) recently. Therefore, in the following section, we introduce a better method for estimating the effect of gradient mismatch problem.

## 4 ESTIMATING THE GRADIENT MISMATCH

### 4.1 GRADIENT OF SMOOTHED LOSS FUNCTION

Since the true gradient of quantized activation network is zero almost everywhere, using the value of the true gradient does not provide useful measure of the gradient mismatch problem. To mitigate the issue with zero true gradient values, Yin et al. (2019) recently introduced a method using the gradient for the expected loss given the distribution of the data to analyze the effect of different STEs. However, the distribution of the dataset is usually unknown and the gradient values of the layers other than the first layer are also zero in multi-layered networks.

In this work, we propose to use the gradient of the *smoothed* loss function for gradient mismatch analysis. Using the simplest rectangular smoothing $s_{\text{rect}}$, the gradient of the smoothed loss function $(s_{\text{rect}} * f)$ along each parameter can be formulated as follows:

$$
\begin{aligned}
\nabla_{\boldsymbol{x}}(s_{\text{rect}} * f) &= \left( \frac{\partial}{\partial x_1} \int_{-\varepsilon}^{\varepsilon} \frac{f(\boldsymbol{x} + s \cdot e^{(1)})}{2\varepsilon} ds, ..., \frac{\partial}{\partial x_n} \int_{-\varepsilon}^{\varepsilon} \frac{f(\boldsymbol{x} + s \cdot e^{(n)})}{2\varepsilon} ds \right) \\
&= \left( \frac{\partial}{\partial x_1} \int_{x_1-\varepsilon}^{x_1+\varepsilon} \frac{f(s, x_2, ..., x_n)}{2\varepsilon} ds, ..., \frac{\partial}{\partial x_n} \int_{x_n-\varepsilon}^{x_n+\varepsilon} \frac{f(x_1, ..., x_{n-1}, s)}{2\varepsilon} ds \right) \\
&= \left( \frac{f(\boldsymbol{x} + \varepsilon \cdot e^{(1)}) - f(\boldsymbol{x} - \varepsilon \cdot e^{(1)})}{2\varepsilon}, ..., \frac{f(\boldsymbol{x} + \varepsilon \cdot e^{(n)}) - f(\boldsymbol{x} - \varepsilon \cdot e^{(n)})}{2\varepsilon} \right) \\
&= \overline{\nabla}_{\varepsilon, \boldsymbol{x}} f
\end{aligned}
\tag{3}
$$

where $e^{(i)}$ refers to standard basis vector with value one at position $i$.

We named $\overline{\nabla}_{\varepsilon, \boldsymbol{x}} f$ as *Coordinate Discrete Gradient (CDG)* since it calculates the difference between discrete points along each parameter dimension. Note that the formula for CDG is very close to the definition of the true gradient which is $\lim_{\varepsilon \to 0} \overline{\nabla}_{\varepsilon, \boldsymbol{x}} f$. While the true gradient indicates the instant rate of change of loss ($\varepsilon \to 0$), CDG is based on the average rate of change in each coordinate. We provide a graphical explanation on CDG in Appendix B.

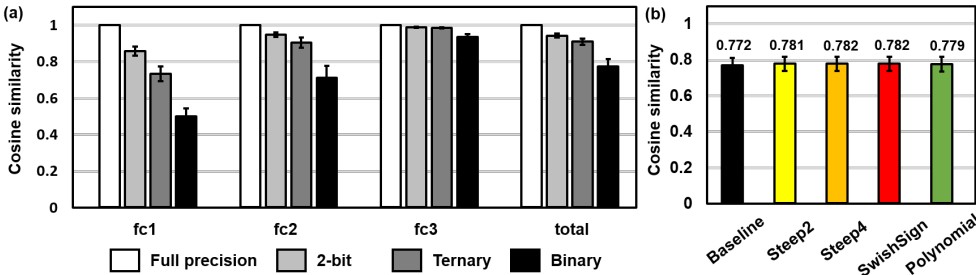

Figure 3: Cosine similarity between true/coarse gradient and CDG in a feed-forward network with 3 hidden layers. (a) Cosine similarity value of each layer when different precision is used for activation. The label *total* refers to the cosine similarity when gradients for each layer are concatenated. (b) Cosine similarity values when sophisticated STEs are used.

Note that the evolutionary strategy-style gradient estimation is another good candidate which can be used for the same purpose as it provides an unbiased gradient estimate for a smoothed function in a similar way (Choromanski et al., 2018). Please refer to Appendix H for more information.

## 4.2 Estimating the gradient mismatch using cosine similarity

Using CDG, we can quantitatively estimate the gradient mismatch. The cosine similarity between CDG and coarse gradient represents how similar the coarse gradient calculated by back-propagation is to CDG, and we can see the effects of 1) different precisions for activation and 2) the sophisticated STEs in the gradient mismatch problem. The detailed algorithm for calculating the cosine similarity between coarse gradient and CDG is described in algorithm 1 in Appendix C. Since the computational burden of calculating CDG is heavy even on GPU, we used a toy model and dataset for experiment. We used a feed-forward neural network with 3 hidden layers. Each hidden layer had 32 neurons and we used Gaussian distribution as the dataset as in Yin et al. (2019).

Figure 3 shows the cosine similarity between CDG and coarse gradient in different cases. First of all, the cosine similarity between CDG and true gradient in full precision case which has smooth loss surface is almost 1 in every layer. This implies that the CDG points to a similar direction to the one which the true gradient points to if the loss surface is not discrete. We also observed that the cosine similarity of the binary activation network was significantly lower than that of higher precision networks and the degradation increases as a layer is further away from the last layer. This is because the gradient mismatch is accumulated as the gradient propagates through layers from the end of network. On the other hand, we found that using sophisticated STEs does not improve the cosine similarity of binary activation networks (Figure 3b), which is consistent with the accuracy results in Figure 2. The results indicate that the cosine similarity between coarse gradient and CDG can provide better correlations with the performance of a model than previous approaches do. In addition, it can be seen that gradient mismatch problem can be reduced more effectively by using higher precision (e.g. ternary or 2-bit) activations than using sophisticated STEs.

We also measured the cosine similarity between the estimated gradient using evolutionary strategies and the coarse gradient. The results show a similar trend to that of the results using CDG (Figure 15 in Appendix H). Most notably, the evolutionary strategy based gradient mismatch assessment also shows that there is a large gap in the cosine similarity between the ternary activation case and the binary activation case with any STEs.

## 5 BinaryDuo: the proposed method

In this section, we introduce a simple yet effective training method called BinaryDuo which couples two binary activations during training to exploit better training characteristics of ternary activation. From the observation in the previous section, we learned that ternary activation functions suffer much less from gradient mismatch problems than binary activation functions with any kind of STE. Therefore, we propose to use such characteristics of ternary activations while training binary acti-

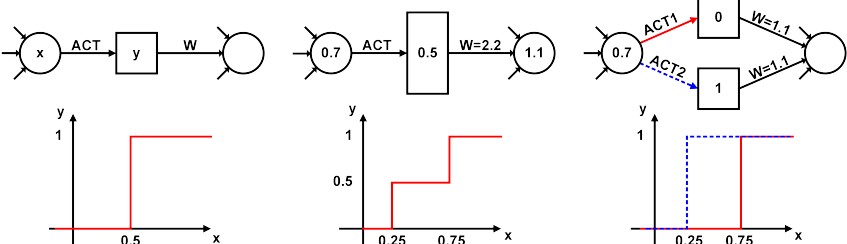

Figure 4: Activation functions of baseline binary (left), ternary (middle) and decoupled binary (right) models. Circle, square and rectangle denote full-precision, binary and ternary data, respectively.

vation networks. The key idea is that a ternary activation function can be represented by two binary activation functions (or thresholding functions).

The BinaryDuo training scheme consists of two stages. In the first stage, a network with ternary activation function is trained. This pretrained model is then deployed to a binary activation network by decoupling a ternary activation into a couple of binary activations, which provides the initialization values of the target binary activation network. In the next stage, the decoupled binary network is fine-tuned to find the better weight parameters. Detailed description of each step is given as follows.

### 5.1 DECOUPLING A TERNARY ACTIVATION TO TWO BINARY ACTIVATIONS

We first elaborate how to decouple a ternary activation to two binary activations. Figure 4 shows three different cases of activation functions. A weighted-sum value (x) goes through activation function depicted in each case. The activation value is then multiplied by a certain weight and contributes to the next weighted-sum. For simplicity, batch-normalization layer (BN) and pooling layer are omitted in the figure. The binary activation function used in the baseline network is shown in the left in Figure 4. The figure in the middle explains how ternary activation function works. The ternary activation function can produce three different values (0, 0.5 or 1) for an output as depicted in the figure. Since the input to the activation function is 0.7 in the example, the output (rectangle) becomes 0.5. The activation is then multiplied by the corresponding weight of 2.2 and contributes 1.1 to the next weighted-sum. The right figure shows how to decouple the ternary activation into two binary activations. The weighted-sum value of 0.7 now passes through two different activation functions (red solid and blue dashed), where each activation function has different threshold values (0.75 and 0.25) to mimic the ternary activation function. The two binary activation values are then multiplied by each weight that is half of the original value. Therefore, they contribute the same value of 1.1 as the ternary case to the next weighted-sum. Note that decoupling the ternary activation does not break the functionality. In other words, computation result does not change after decoupling.

As mentioned earlier, the decoupled binary activation function layer should have a threshold of 0.25 in half and 0.75 in the other half to mimic the ternary activation function. In practice, instead of using different threshold values in a layer, we shift the bias of BN layer which comes right before the activation function layer. The ternary activation function can be expressed as follows:

$$y_t = f_t(\text{BN}(\Sigma, \gamma, \beta)). \tag{4}$$

Here, $f_t$ and $y_t$ are ternary activation function and its output as shown in the middle of Figure 4. $\text{BN}(x, \gamma, \beta)$ denotes BN function with input $x$, weight $\gamma$ and bias $\beta$. Similarly, two binary activation functions after decoupling can be expressed as Eq. 5.

$$y_{b1} = f_b(\text{BN}(\Sigma, \gamma, \beta + 0.25)), \quad y_{b2} = f_b(\text{BN}(\Sigma, \gamma, \beta - 0.25)) \tag{5}$$

Again, $f_b$ and $y_b$ are binary activation function and its output as shown in the left of Figure 4. While $\gamma$ of BN is copied from the ternary activation case, $\beta$ is modified during decoupling. When a ternary activation function is decoupled following Eq. 5, $y_t = (y_{b1} + y_{b2})/2$ becomes valid, enabling exact same computation after decoupling. Please note that computations for BN and the binary activation can be merged to a thresholding function. Therefore, modulating the BN bias with different values as in Eq. 5 does not incur additional overhead at inference stage.

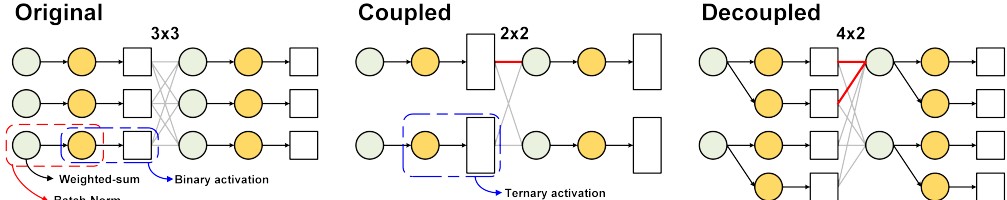

Figure 5: Example model architectures of baseline, coupled ternary model and decoupled binary model with specific width numbers. Two weights highlighted in the decoupled model are half the value of the weight highlighted in the coupled model.

## 5.2 LAYER WIDTH RECONFIGURATION

As shown in Figure 4, the number of weights is doubled after decoupling. Therefore, to match the parameter size of the decoupled model and the baseline model, we must reduce the size of the coupled ternary model. Figure 5 shows an example how we choose the network size. Each diagram shows BN-activation (ACT)-fully connected (FC)-BN-ACT layers in order from the left. In case the baseline model has a 3x3 configuration for the FC layer, the layer has a total of nine weights. In this case, we train the coupled ternary model with 2x2 configuration which has a total of 4 weights. The third diagram shows the corresponding decoupled binary model. As explained in Eq. 5, a weighted-sum value generates two BN outputs with different BN biases. Since decoupling doubles the number of weights in the model, the decoupled model has 8 weights which is still less than the number of weights in the baseline model. To make sure that the number of total weights in the decoupled binary model does not exceed that of the baseline model, we use a simple formula, $N_{\text{coupled}} = \lfloor N_{\text{baseline}}/\sqrt{2} \rfloor$, where N denotes the number of neurons (or channels in convolution).

## 5.3 FINE-TUNING THE DECOUPLED MODEL

After decoupling, two weights derived from the same weight in the coupled model have the same value (1.1 in the example) since they were tied in the ternary activation case. In the fine-tuning stage, we update each weight independently so that each of them can find a better value since the two weights do not need to share the same value anymore. As mentioned earlier, decoupling itself does not change the computation result, so the same accuracy as that of the coupled ternary model is achieved before fine-tuning and experimental results show that the fine-tuning increases the accuracy even further. Detailed experimental results will be discussed in the next section. Since the initial state before fine-tuning can be already close to a good local minimum when the model is decoupled, we used much smaller learning rate for fine-tuning.

## 6 EXPERIMENTAL RESULTS

### 6.1 VGG-7 ON CIFAR-10

We evaluate the proposed training method in two steps. First, we validate the effectiveness of the BinaryDuo scheme with VGG-7 on CIFAR-10 dataset. We used the same setting with the experiment introduced in Figure 1. We first trained the coupled ternary model, then we decoupled the trained coupled ternary model as explained in section 5.1. The training results with various training schemes and activation precision are summarized in Figure 6. The three contour plots show the hyper parameter search results for each case. Please refer to Appendix A for more details on experimental setup. First of all, note that the best test accuracy of the coupled ternary model is 89.69% which is already higher than 89.07% accuracy of the baseline binary model. In other words, using ternary activation and cutting the model size in half improved the performance of the network. By fine-tuning the decoupled binary model, we could achieve 90.44% test accuracy, which is 1.37% higher than the baseline result. It can be observed that the steep drop of accuracy from 2b activation case to 1b activation case is significantly reduced with our scheme.

Even though the decoupled binary model has the same (or less) number of parameters and computing cost than the baseline model, someone may wonder whether the performance improvement might

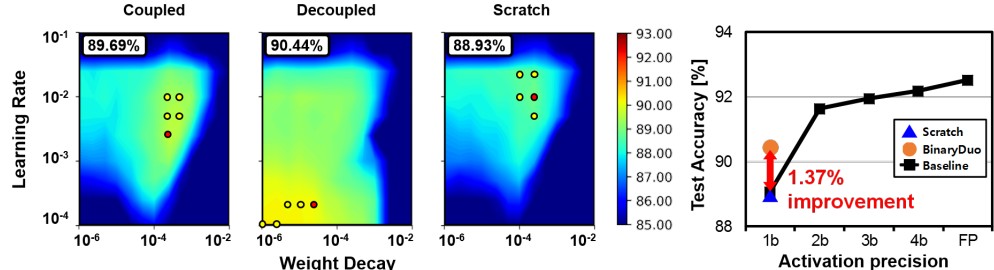

Figure 6: Training results of VGG-7 on CIFAR-10 dataset in the order of coupled ternary model, decoupled binary model and decoupled binary model trained from scratch (left). 5 circles in each plot represent the top 5 test accuracy points and the red circle is for the best result. The best accuracy result is shown at the top left corner of each contour plot. Test accuracy of various models with different activation precision and training schemes (right).

Table 2: Classification accuracy of BNN in various schemes.

| | AlexNet | | | | ResNet-18 | | | |
|---|---|---|---|---|---|---|---|---|
| | Accuracy | | Params | Comp. | Accuracy | | Params | Comp. |
| Network | Top-1 | Top-5 | (Mbit) | (FLOP) | Top-1 | Top-5 | (Mbit) | (FLOP) |
| BNN | 41.8 | 67.1 | 62.3 | 82.3M | - | - | - | - |
| XNOR-Net | 44.2 | 69.2 | 191 | 126M | 51.2 | 73.2 | 33.3 | 164M |
| BNN+ | 46.1 | 75.7 | 191 | 126M | 53.0 | 72.6 | 33.3 | 164M |
| Yang et al. (2019) | 47.9 | 72.5 | 191 | 126M | 53.6 | 75.3 | 33.3 | 164M |
| Bi-Real-Net[†] | - | - | - | - | 56.4 | 79.5 | 33.3 | 164M |
| WRPN 2x | 48.3 | - | 498 | 283M | - | - | - | - |
| ABC-Net | - | - | - | - | 42.7 | 67.6 | 33.3 | 164M |
| Group-Net | - | - | - | - | 55.6 | 78.6 | 33.3 | 164M |
| CBCN[†] | - | - | - | - | 61.4 | 82.8 | 33.3 | 656M |
| Wang et al. (2019)[†] | - | - | - | - | 59.9 | 84.2 | 36.7 | 183M |
| Ding et al. (2019) | 47.8 | 71.5 | 191 | 126M | - | - | - | - |
| Bulat et al. (2019) | - | - | - | - | 55.6 | 78.5 | 33.3 | 164M |
| Gu et al. (2019)[†] | - | - | - | - | 57.3 | 80.0 | 33.7 | 164M |
| Bethge et al. (2019)[†] | - | - | - | - | 57.7 | 80.0 | 33.3 | 164M |
| **BinaryDuo** | **52.7** | **76.0** | **189** | **119M** | **60.4** | **82.3** | **31.9** | **164M** |
| **BinaryDuo(+sc)[†]** | - | - | - | - | **60.9** | **82.6** | **31.9** | **164M** |

[†] These schemes use ResNet models with additional shortcut proposed by Liu et al. (2018).

be an outcome from the modified network topology. Therefore, we conducted another experiment to rule out the possibility. We trained the decoupled binary model from scratch without using the pretrained ternary model. The result is shown in the third contour plot of Figure 6. The best accuracy was 88.93% which is lower than the baseline result. Hence, we conclude that the proposed training scheme plays a major role for improving the performance of binary activation network.

## 6.2 BNN ON IMAGENET

We also compare BinaryDuo to other state-of-the-art BNN training methods. We used the BNN version of AlexNet (Krizhevsky et al., 2012) and ResNet-18 (He et al., 2016) on ImageNet (Deng et al., 2009) dataset for comparison. Details on training conditions are in Appendix D.

Table 2 shows the validation accuracy of BNN in various schemes. For fair comparison, we also report the model parameter size and computing cost at inference stage of each scheme. Computing cost (FLoating point OPerations) was calculated following the method used in Liu et al. (2018) and Wang et al. (2019). For ResNet-18 benchmark, we report two results with and without additional

shortcut as proposed in Liu et al. (2018). Regardless of the existence of additional shortcut path, BinaryDuo outperforms state-of-the-art results with same level of parameter size and computing cost for both networks. BinaryDuo improved the top-1 validation accuracy by 4.8% for AlexNet and 3.2% for ResNet-18 compared to state-of-the-art results. CBCN (Liu et al., 2019) achieved slightly higher top-1 accuracy than BinaryDuo but it requires more computing cost than BinaryDuo. CBCN uses similar amount of parameters and computing cost to that of the multi-bit network which uses 4-bit activations and 1-bit weights.

## 7 CONCLUSION AND FUTURE WORK

In this work, we used the gradient of smoothed loss function for better gradient mismatch estimation. The cosine similarity between the gradient of the smoothed loss function and the coarse gradient shows higher correlation with the performance of a model than previous approaches. Based on the analysis, we proposed BinaryDuo which is a new training scheme for binary activation network. The proposed training scheme improves top-1 accuracy of BNN version of AlexNet and ResNet-18 for 4.8% and 3.2% compared to state-of-the-art results, respectively, by minimizing gradient mismatch problem during training. In the future, we plan to explore several techniques proposed for multi-bit network to improve the performance of BinaryDuo and to apply the proposed training scheme on other tasks such as speech recognition and object detection.

ACKNOWLEDGMENTS

This work was supported by the MSIT(Ministry of Science and ICT), Korea, under the ICT Consilience Creative program(IITP-2019-2011-1-00783) supervised by the IITP(Institute for Information & communications Technology Planning & Evaluation). This work was also supported by the MOTIE(Ministry of Trade, Industry & Energy (10067764) and KSRC(Korea Semiconductor Research Consortium) support program for the development of the future semiconductor device.

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

# APPENDIX

## A TRAINING CONDITIONS AND RESULTS FOR VGG-7 ON CIFAR-10

For the experiments in section 3.1, we trained various VGG-7 models with different activation precisions and width of layers on CIFAR-10 dataset. Width of a layer indicates number of neurons in fully-connected layers or number of channels in convolution layers. The baseline VGG-7 model which we used consists of 4 convolution layers with 64, 64, 128 and 128 channels and 3 dense layers with 512, 512 and 10 output neurons. Max pooling layer was used to reduce the spatial domain size after the second, third and fourth convolution layers. Full precision weights were used for all cases while the activation function layers except the last one were modified following the activation precision.

Each model was trained for 200 epochs with learning rate scaling by 0.1 at 120th and 160th epochs. We initialized models following He et al. (2015) and the mini-batch size was fixed to 256. We used Adam optimizer with decoupled weight decay (Loshchilov & Hutter, 2019) for easier hyper-parameter search. We tried 13 different exponentially increasing weight decay values and 10 different exponentially increasing initial learning rates. Test accuracy from at least two runs were averaged for each combination of weight decay and initial learning rate. Figure 7 shows the test accuracy of benchmark networks with various activation precisions.

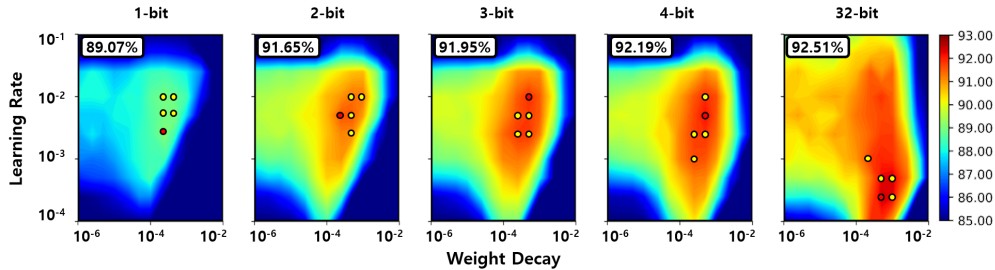

Figure 7: Training results of VGG-7 on CIFAR-10 dataset. 5 circles represent top 5 test accuracy points and the red circle is for the best result. The best test accuracy in each contour plot is shown on top of the contour plot.

## B GRAPHICAL EXPLANATION ON CDG

In this section, we elaborate the difference between true gradient and CDG using bivariate loss function. Figure 8 shows graphical explanation of the concept of CDG. Each plot in the figure shows how to calculate the true gradient and CDG, respectively. The green contour plot in each case represents the given loss surface ($\mathcal{L}$) which is a function of $w_1$ and $w_2$. Red and blue lines in the edge shows the cross-section of the loss surface in direction of the dashed lines. True gradient at the given point can be derived by calculating the slope of loss surface in each direction. Similarly, CDG can be obtained by calculating how much the loss changes when we move to each direction by given step size ($\varepsilon$). Then, by calculating the cosine similarity between the two vectors, we evaluate how similar the directions of two vectors are.

## C COSINE SIMILARITY BETWEEN CDG AND COARSE GRADIENT

### C.1 STEP SIZE OF CDG

When calculating the CDG, the size of $\varepsilon$ in Eq. 3 has to be carefully chosen. First, the $\varepsilon$ has to be small enough to minimize the approximation error from non-linearity of the loss surface. At the same time, it has to be large enough to ignore the noise from rough and uneven stepwise loss surface caused by quantized activation. In addition, the optimal size of $\varepsilon$ depends on the number of samples used for loss calculation. If the number of samples used for the loss calculation is small, the larger

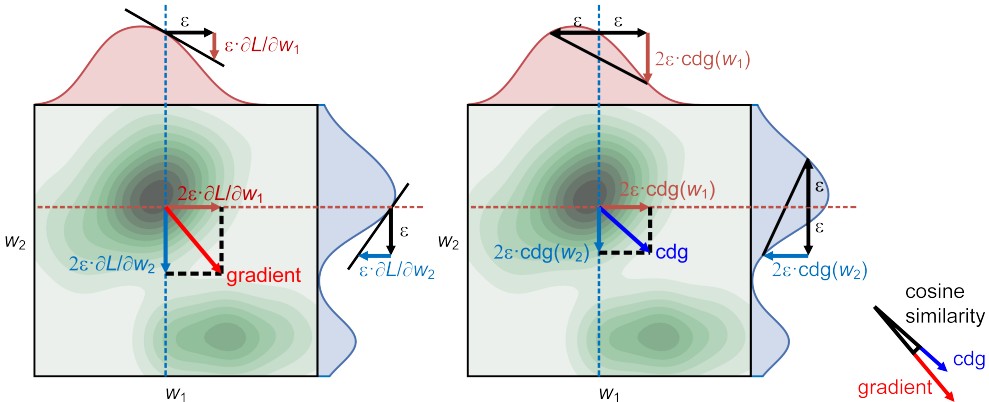

Figure 8: Graphical explanation of CDG using bivariate loss function.

size of $\varepsilon$ is preferred to minimize the noise from the rough and uneven loss surface (Figure 9a,b). On the other hand, if the number of samples used for the loss calculation is large, the smaller size of $\varepsilon$ is preferred to better estimate the true gradient (Figure 9c,d). In our experiment, we used one million samples and $\varepsilon$ of 0.001 which are in a similar range to the dataset size and learning rate.

Although the absolute values of cosine similarity may depend on the choice of $\varepsilon$, the overall trend over different activation bit precision is maintained regardless of the $\varepsilon$ value. Figure 10 shows how cosine similarity changes as the step size varies from 1e-4 to 1e-2. The gap between ternary activation and binary activation was clearly shown in every layer regardless of the step size (Figure 10a,c,e and g). In addition, binary activation networks with any STE showed similar cosine similarity values regardless of the epsilon value (Figure 10b,d,f and h).

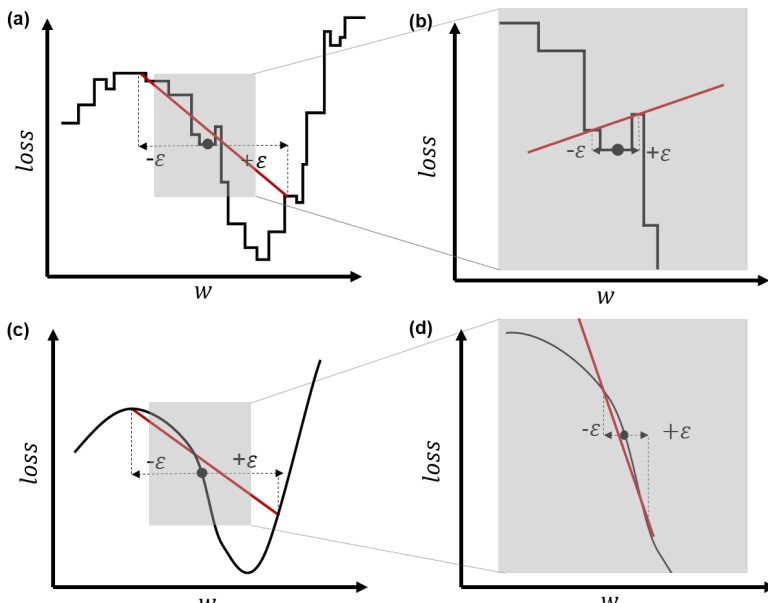

Figure 9: The examples of loss curve and coordinate discrete gradient in a single dimension with different scale of $\varepsilon$ and the different number of samples used to calculate the loss curve. The number of samples used is small (a and b) and large (c and d). The size of $\varepsilon$ is large (a and c) and small (b and d).

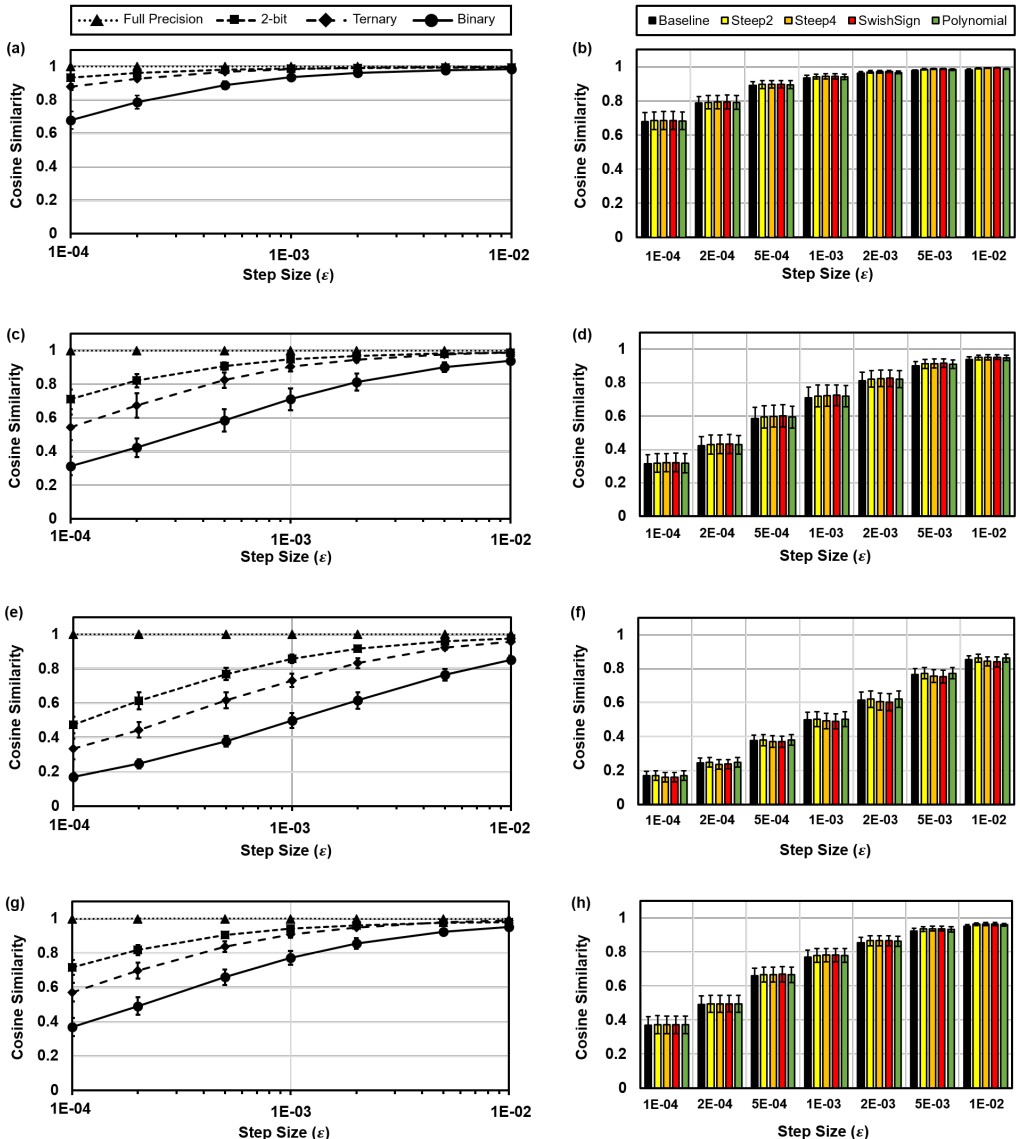

Figure 10: Effect of step size ($\varepsilon$) on cosine similarity between CDG and coarse gradient with different precision (a,c,e and g) and with different sophisticated STEs (b,d,f and h). Cosine similarity trend of each layer (fc1, fc2, fc3 and total) is shown in each row from top to bottom. It can be observed that there is a large difference in cosine similarity between ternary activation and binary activation regardless of the step size. In addition, all BNNs have similar cosine similarity values regardless of STE choices and step size.

## C.2 ALGORITHM FOR CALCULATING COSINE SIMILARITY

Algorithm 1 explains how we calculated the cosine similarity between coarse gradient and CDG in Section 4. First, we generate the dataset by sampling from the standard normal distribution in i.i.d manner. Then we initialize the weight for target model and evaluating model. The target model is used for calculating the target value for computing the loss value of evaluating model. The coarse gradient of each weight matrix is calculated by back-propagation with STE. For calculating the CDG of a single element, we evaluate the loss at two parameter points $\boldsymbol{W}_i + \varepsilon \cdot e^{(j,k)}$ and $\boldsymbol{W}_i - \varepsilon \cdot e^{(j,k)}$, which are nearby points of current parameter point along the coordinate of the selected element. After evaluating the coarse gradient and CDG, we can estimate the gradient mismatch by calculating cosine similarity between those two.

---

**Algorithm 1** Calculating the cosine similarity between the coordinate discrete gradient and the coarse gradient with respect to $\varepsilon, f$

---

Generate $\boldsymbol{X} \in \mathbb{R}^{m \times n}$ from standard normal distribution $\mathcal{N}(0, 1)$

Initialize $\boldsymbol{W}_1, \ldots, \boldsymbol{W}_{l-1}, \boldsymbol{W}_1^*, \ldots, \boldsymbol{W}_{l-1}^* \in \mathbb{R}^{m \times m}, \boldsymbol{W}_l, \boldsymbol{W}_l^* \in \mathbb{R}^m$

Define the evaluating model $F(\boldsymbol{x}) = \boldsymbol{W}_l(f(\ldots, \boldsymbol{W}_2 f(\boldsymbol{W}_1 \boldsymbol{x})))$

Define the target model $F^*(\mathbf{x}) = \boldsymbol{W}_l^*(f(\ldots, \boldsymbol{W}_2^* f(\boldsymbol{W}_1^* \boldsymbol{x})))$

$\mathcal{L} \leftarrow \frac{1}{2n} \sum_{t=1}^{n} (F(\boldsymbol{X}_{:,t}) - F^*(\boldsymbol{X}_{:,t}))^2$

**for** $i = 1, 2, \ldots, l$ **do**

    # Calculate the coarse gradient using STE

    $\mathbf{C}_{i,:,:} \leftarrow \tilde{\nabla}_{\boldsymbol{W}_i} \mathcal{L}$

    # Calculate the coordinate discrete gradient

    **for** $j = 1, 2, \ldots, m$ **do**

        **for** $k = 1, 2, \ldots, m$ **do**

            $\mathcal{L}_+ = \frac{1}{2n} \sum_{t=1}^{n} (F(\boldsymbol{X}_{:,t}; \boldsymbol{W}_i + \varepsilon \cdot e^{(j,k)}) - F^*(\boldsymbol{X}_{:,t}))^2$

            $\mathcal{L}_- = \frac{1}{2n} \sum_{t=1}^{n} (F(\boldsymbol{X}_{:,t}; \boldsymbol{W}_i - \varepsilon \cdot e^{(j,k)}) - F^*(\boldsymbol{X}_{:,t}))^2$

            $\mathbf{D}_{i,j,k} \leftarrow (\mathcal{L}_+ - \mathcal{L}_-)/2 \cdot \varepsilon$

        **end for**

    **end for**

    # Calculate the cosine similarity

    $s_i \leftarrow \frac{\mathbf{C}_{i,:,:} \cdot \mathbf{D}_{i,:,:}}{|\mathbf{C}_{i,:,:}||\mathbf{D}_{i,:,:}|}$

**end for**

$s_{total} \leftarrow \frac{\mathbf{C} \cdot \mathbf{D}}{|\mathbf{C}||\mathbf{D}|}$

**return** $\{s_1, s_2, \ldots, s_l, s_{total}\}$

---

## D    IMAGENET TRAINING CONDITIONS AND TRAINING CURVE

For comparison with state-of-the-art BNN results, we used ImageNet (Deng et al., 2009) dataset which has 1.28M images for training set and 50K images for validation set. We initialized the coupled ternary model following He et al. (2015) for all cases and the mini-batch size was fixed to 256. Weights are also binarized following Rastegari et al. (2016) which uses layerwise scaling factor. We also clipped weights so that weights are always between -1 and +1. For AlexNet, we used Batch-normalization layer instead of local response normalization layer. Since binarization of weights and activations serves as strong regularizer, we used 0.1 for dropout ratio in AlexNet. The maximum training epoch was 100 for AlexNet and we used a learning rate starting at 1e-3 and decreasing by 1/10 at 41, 71 and 91 epochs. Weight decay of 2e-5 was used and Adam optimizer was used. When fine-tuning the decoupled model, we trained the model for 40 epochs. Initial learning rate was set to 2e-5 and multiplied by 0.1 at 16, 26 and 36 epochs. Weight decay was also lowered to 1e-6 when fine-tuning. For ResNet-18, we used shortcut type of 'B' as in other BNN papers. We used full precision shortcut path since its overhead is negligible as described in (Rastegari et al., 2016; Liu et al., 2018). Maximum number of training epoch was 120 and we did not use weight decay for ResNet-18. Initial learning rate was set to 5e-3 and multiplied by 0.1 at 71, 91 and 111 epochs. For fine-tuning, we also trained the model for 40 epochs. Learning rate was set to 2e-3 and decayed at 21, 31 and 36 epochs. Training curves for both AlexNet and ResNet-18 are shown in Figure 11.

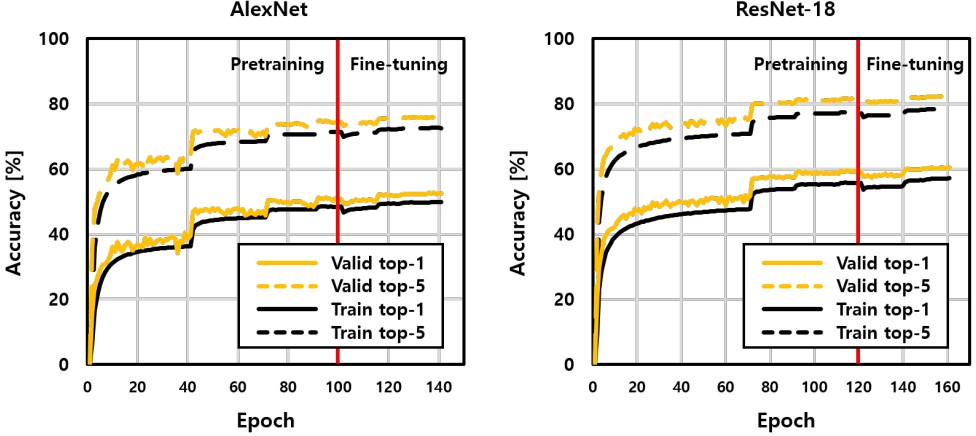

Figure 11: Training curves for AlexNet and ResNet-18 with BinaryDuo training scheme.

## E    COUPLED MODEL WITH HIGHER PRECISION

Based on the observation that the binary activation function suffers more from gradient mismatch problem than ternary activation function, we used a model with ternary activation function for coupled pretrained model. However, the proposed training scheme can also use the higher precision models as the pretrained model in the same manner. For example, a model with 2-bit activation function, which can be mimicked by three binary activations, can be used as the pretrained model. In this case, however, the model size of the 2-bit pretrained model should be even smaller than that of the ternary pretrained model since decoupling the 2-bit activation function increases the number of weights $3\times$. We evaluated such case on the same benchmark which was also used in Section 6.1.

Figure 12 shows the training results of the coupled 2-bit model and the corresponding decoupled binary model. Compared to the coupled ternary model reported in Section 6.1, the coupled 2-bit model achieved similar test accuracy (89.61%) which is also higher than baseline result. However, the performance of the binary model which was decoupled from the 2-bit model could not achieve a comparable performance to the value of the network decoupled from the ternary network which was reported in Section 6.1. We further trained with even lower learning rate and weight decay but we could not achieve better performance. This result was interesting because decoupling 2-bit model increases the number of weights in a model 3 times while decoupling ternary model doubles

the number of weights. We suspect that the cause of this phenomenon is that the local minimum value that was found by the pretrained model becomes less useful when the model is more deformed through decoupling.

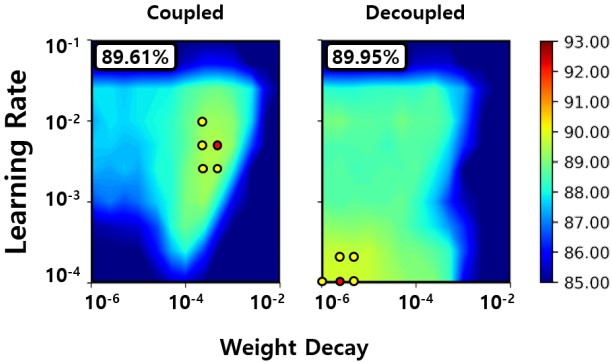

Figure 12: Training result of coupled 2-bit model and the corresponding decoupled model.

## F BINARYDUO WITHOUT MODEL MODIFICATION

In this section, we introduce a variation of BinaryDuo that can keep the baseline model architecture even after the decoupling. While the proposed method in Section 5 uses the half-sized ternary model as the pretrained coupled model, this approach uses quarter-sized ternary model as the pretrained model. Let us call this approach as BinaryDuo-Q. BinaryDuo-Q quadruples the number of weights of a model when decoupling as shown in Figure 13. Suppose that the original baseline network has 4x4 configuration. BinaryDuo-Q reduces the width of all layers by half to make coupled ternary model. In the example case, the coupled model has 2x2 configuration as shown in the left side of Figure 13. When decoupling, one weight from the coupled model produces four weights in the decoupled model. While the decoupled model of BinaryDuo-Q has the same architecture as the original baseline model, the coupled ternary model has half number of parameters of the coupled ternary model used for BinaryDuo in Section 5. Figure 14 shows the training results of BinaryDuo-Q. As expected, the coupled ternary model with the quarter size of baseline model showed worse performance than the half-sized coupled model due to the less number of parameters. As a result, even though decoupling the quarter-size model improves the performance by 1.67%, the best test accuracy of the decoupled model using BinaryDuo-Q was 89.72% which is lower than that using BinaryDuo. Nevertheless, note that the result of BinaryDuo-Q is still better than baseline result.

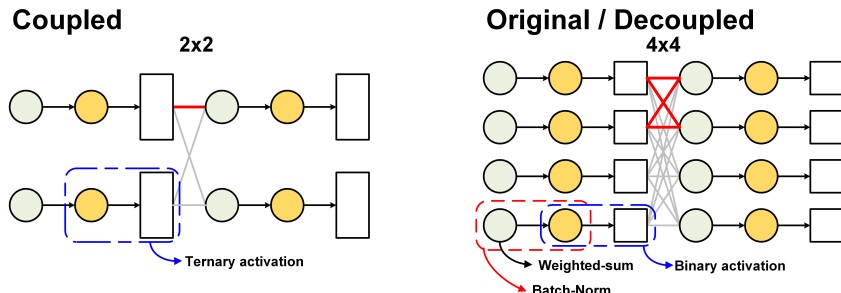

Figure 13: Example model architectures of coupled ternary and decoupled binary model when using BinaryDuo-Q. Coupled ternary model with 2x2 configuration is decoupled to binary model with 4x4 configuration.

## G BETTER MULTI-BIT TRAINING

Recently, several works proposed to learn the interval and range of quantized activation function for higher accuracy (Jung et al., 2019; Choi et al., 2018; 2019; Zhang et al., 2018). Unfortunately,

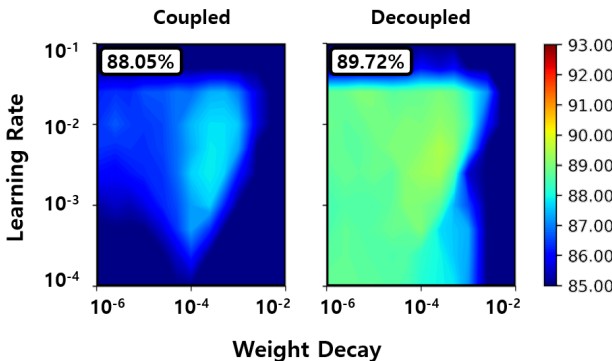

Figure 14: Training results of coupled ternary model and decoupled binary model when using BinaryDuo-Q.

these techniques were not effective on binary activation networks. However, we use the multi-bit network as a pretrained model for binary activation network, and hence there is a chance of accuracy improvement using such interval learning methods. In the future, we plan to explore such techniques to improve the performance of BinaryDuo.

## H ESTIMATING THE GRADIENT MISMATCH USING EVOLUTIONARY STRATEGY-STYLE GRADIENT ESTIMATOR

As mentioned in Section 4, evolutionary strategy-style gradient estimator (ESG) can also be used for estimating the gradient mismatch similar to CDG because both CDG and ESG provide the information of the gradient of smoothed loss function. In particular, among many evolutionary strategy-style gradient estimators, the antithetic evolutionary strategy gradient estimator has a similar functional form with CDG as shown below.

$$\hat{\nabla}_N f_\sigma(\boldsymbol{x}) = \frac{1}{2N\sigma} \sum_{i=1}^{N} (f(\boldsymbol{x} + \sigma\varepsilon_i)\varepsilon_i - f(\boldsymbol{x} - \sigma\varepsilon_i)\varepsilon_i) \tag{6}$$

where $(\varepsilon_i)_{i=1}^N \overset{\text{iid}}{\sim} \mathcal{N}(0, I)$. The cosine similarity between ESG and the coarse gradient has been computed for analysis (Figure 15). Since $\sigma$ in Eq. 6 controls the sharpness of the smoothing kernel as $\varepsilon$ does in CDG, we calculated the cosine similarity between ESG and the coarse gradient with various size of $\sigma$. For fair comparison with CDG in terms of computational cost, we used 1024 samples for gradient estimation ($N$=1024 in Eq. 6).

Note that findings from Figure 3 and Figure 10 were that 1) a large gap exists between the cosine similarities in ternary activation case and binary activation case and 2) use of sophisticated STEs does not have a significant effect on cosine similarity. Those two trends remained the same in the experiment using ESG as shown in Figure 15.

It is also worthwhile to note that the cosine similarity between the true gradient (full precision case) and ESG is around 0.7 (Figure 15) while the cosine similarity between the true gradient and CDG is close to 1 (Figure 10). It may imply that CDG can be considered as more reliable metric than ESG given the same amount of computational cost. However, in-depth study is needed to assess the advantage and disadvantage of CDG over ESG more thoroughly.

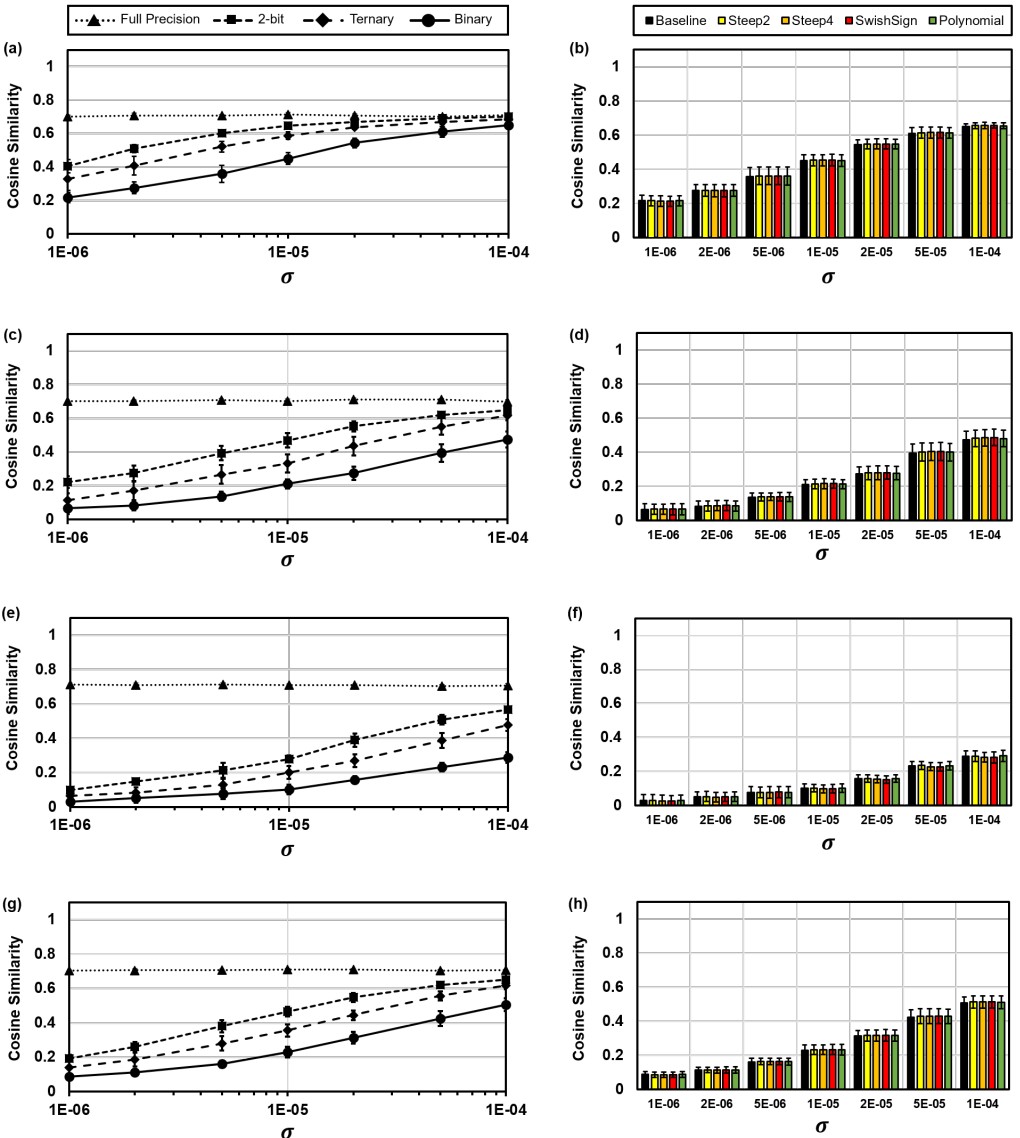

Figure 15: Effect of $\sigma$ on cosine similarity between ESG and coarse gradient with different precision (a,c,e and g) and with different sophisticated STEs (b,d,f and h). Cosine similarity trend of each layer (fc1, fc2, fc3 and total) is shown in each row from top to bottom.

