# OpenReview forum: "BinaryDuo: Reducing Gradient Mismatch in Binary Activation Network by Coupling Binary Activations"
_ICLR.cc/2020/Conference — Accept (Poster)_

### Official Review · AnonReviewer1 · 2019-10-23
**Official Blind Review #1**

**Rating:** 6

**Review:**

This paper studies activation quantization in deep networks. The authors first compare the coordinate discrete gradient and those obtained by various kinds of straight-through estimators, and found 1-bit activation networks have much poorer gradient estimation than 2-bit ones. Thus they speculate that this explains the poorer performance of 1-bit activation networks than 2-bit ones. To utilize higher precision of activation, the authors then propose to decouple a ternary activation into two binary ones, and achieve competitive results on typical image classification data sets CIFAR-10 and ImageNet.

The paper is overall well-written and easy to follow. The decoupling method is simple and straightforward. The experiments are also well conducted. One main concern is that since the computation of the decoupled binary model and the coupled ternary model are the same, why does the decoupled binary model can finally to tuned to perform better than the original ternary model? Is there any intuition or theoretical explanation? Yet another concern is that ternary activation basically can be viewed as binary+sparse activations, can it be even more computationally cheaper than the decoupled binary activation?

Question:
1. One line below eq (2), does STE mean the estimated gradient? How can the difference be calculated based on different things (i.e., activations and gradients)?

**Experience Assessment:**

I have published one or two papers in this area.

**Review Assessment: Checking Correctness Of Derivations And Theory:**

I assessed the sensibility of the derivations and theory.

**Review Assessment: Checking Correctness Of Experiments:**

I assessed the sensibility of the experiments.

**Review Assessment: Thoroughness In Paper Reading:**

I read the paper at least twice and used my best judgement in assessing the paper.

---

> ### Author Response · Authors · 2019-11-12
> **Response to Reviewer #1 (part 2)**
>
>
> Q3: One line below eq (2), does STE mean the estimated gradient? How can the difference be calculated based on different things (i.e., activations and gradients)?
>
> A3: After reading reviewer’s question, we noticed that there might be a confusion over the terminology ‘STE’. Therefore, we updated our manuscript to clarify our intention as follows.
> We use the term ‘STE’ to indicate the derivative of the approximation of the binary activation function used at backward pass. For example, derivative of HardTanh function was used as STE in Courbariaux et al. (2016). In Figure 2, g’(x) is for STE.
> We call the presumed activation function which is used at backward pass as differentiable approximation of binary activation function. For example, HardTanh or SwishSign is one of the differentiable approximations of the binary activation function. In Figure 2 and Eq. 2, g(x) is for the differentiable approximation of the binary activation function.
> Therefore, STE (g’(x)) is the derivative of the differentiable approximation of binary activation function (g(x)).
> The cumulative difference in Eq. 2 was used to measure the difference between the actual binary activation function and its differentiable approximation. Yellow area in Fig.2 will help understanding Eq.2 graphically.
>
> We thank the reviewer for pointing this out and helping us to improve our manuscript for better understanding.

---

> ### Author Response · Authors · 2019-11-12
> **Response to Reviewer #1 (part 1)**
>
> Thank you very much for your constructive comments. We could improve the quality of our manuscript thanks to your comments. Here are our responses.
>
>
> Q1: One main concern is that since the computation of the decoupled binary model and the coupled ternary model are the same, why does the decoupled binary model can finally to tuned to perform better than the original ternary model? Is there any intuition or theoretical explanation?
>
> A1: When a ternary model is decoupled, a weight is divided into two identical weights as shown in Figure 4 (right). Before fine-tuning starts, the two weights have the same value, which is half of the original weight value. However, once fine-tuning begins, the two weights are allowed to have different values with backpropagation because they are connected to separate neurons with different threshold values after decoupling. Therefore, it becomes possible to find better minima by letting them independently be updated during the fine-training process.
>
>
> Q2: Yet another concern is that ternary activation basically can be viewed as binary+sparse activations, can it be even more computationally cheaper than the decoupled binary activation?
>
> A2: As reviewer rightfully suggested, a ternary activation can be viewed as a binary+sparse activations and bitwise binary operations can be applied to compute it. However, we think that such an approach is more expensive to compute than the decoupled binary activation. Let us elaborate the detail process to assess the computing overhead.
>
> Let $Y$ be a dot-product result of a ternary input activation vector ($X$) and a binary weight vector ($W$) with length of n each. $X \in \{-1,0,1\}$ has to be encoded to a 2-bit binary number: for example, $\{10, 00, 11\}$.
> Let us call the higher bit of the 2-bit input which indicates whether the number is zero or not as $X^{MSB}$, and the other bit as $X^{LSB}$. Computations for ternary activation consist of 3 steps as follows.
>
> Step 1: To utilize bitwise operation for computing ternary activation, we first need to count the number of inputs that are not zero.
>     (1)    $POPCNT(X^{MSB}) = m$
>
> Step 2: Then, we need to mask zero inputs and pack non-zero inputs which will participate in bitwise computation.
>     (2)    $X^{LSB} \in {0,1}^n ⇒ X^{\emptyset,LSB} \in {0,1}^m, W ⇒ W^{\emptyset}$
> To pair the non-zero value inputs with corresponding weights, index information for the non-zero values needs to be kept. Note that the number of bits to store index information is not negligible compared to the bits required to store the ternary input values. In addition, it takes substantial time to find the matching pair of non-zero input and corresponding weight.
>
> Step 3: Lastly, with XNOR_POPCNT operation of $X^{\emptyset,LSB}$ and $W^{\emptyset}$, we can derive the desired output Y as follows.
>     (3)    $Y = 2*{\text{XNOR_POPCNT}(X^{\emptyset,LSB}, W^{\emptyset})}-m$
>
> In summary, computing ternary activation with bitwise operations requires the extra index information for 0-input value because 0-input values need to be excluded from the bitwise operation. While the overhead of indexing process is manageable in high-bit precision case, it becomes non-negligible when input has 1-bit or 2-bit precision since the amount of extra index information becomes comparable with the amount of original input data. Therefore, we believe that computing ternary activation as binary+sparse activations is computationally more expensive than the decoupled binary activation.
>
> On the other hand, the cost of conventional computing for ternary activation is actually lower than the cost of computing ternary activation which is regarded as binary+sparse activations.
> In fact, the cost of conventional computing for ternary activation is computationally same as that of computing decoupled binary activation. In this case, the ternary input $X \in \{-1, 0, 1\}$ may be encoded to a 2-bit binary number $\{00, 01, 11\}$, in which the number of `1's indicates the relative order of the numbers. With such encoding, we can derive the desired output Y as follows.
> $Y = 2*{\text{XNOR_POPCNT}(X^{MSB},W)+\text{XNOR_POPCNT}(X^{LSB},W)}-n$
>
> In this case, the cost of computing ternary activation is twice as that of computing binary activation (XNOR_POPCNT). Since decoupling basically splits each ternary activation into two separate binary activations, compute cost of coupled ternary model and that of decoupled binary model are the same. Therefore, we think that cost of computing ternary activation is not cheaper than computing corresponding decoupled binary activation.
> Meanwhile, the accuracy of the decoupled binary model is higher than that of the coupled ternary model after fine-tuning. In conclusion, the decoupled binary model can achieve higher accuracy with the same amount of compute cost as the coupled ternary model.

---

### Official Review · AnonReviewer4 · 2019-10-25
**Official Blind Review #4**

**Rating:** 6

**Review:**

This paper proposes a new measure of gradient mismatch for training binary networks, and additionally proposes a method for getting better performance out of binary networks by initializing them to behave like a ternary network.

I found the new measure of gradient deviation fairly underdeveloped, and I suspect the method of converting ternary activations into binary activations works for a different reason than that proposed by the authors.

There were English language issues that somewhat reduced clarity, though the intended meaning was always understandable.

Detailed comments:

"Binary Neural Network (BNN) has been gaining interest thanks to its computing cost reduction and memory saving." --> "Binary Neural Networks (BNNs) have been garnering interest thanks to their compute cost reduction and memory savings." (will stop making English language corrections from here on)

"Therefore, we argue that the sharp accuracy drop for the binary activation stems from the inefficient training method, not the capacity of the model."
This could also be due to poor initialization in the binary case. e.g., it might make sense to initialize the binary network with bias=-0.5, so that the nonlinearity has a kink at pre-activation=0, rather than pre-activation=0.5.

"Unfortunately, it is not possible to measure the amount of gradient mismatch directly because the
true gradient of a quantized activation function is zero almost everywhere. " It *is* possible to measure the mismatch to the true gradient exactly. One could even train using the true gradient. It's just that the true gradient is useless.

Fig 1b -- this is a nice baseline.

"the steepest descent direction, which is the direction toward the point with the smallest loss at given distance"
This is not the usual definition of steepest descent direction. If you're going to redefine this, should do so mathematically and precisely (for instance, you are going to run into trouble with the word "distance", since your coordinate discrete gradient more closely resembles an L\infty-ball perturbation, rather than an L2-ball perturbation.

eq. 3:
Note that this equation is equivalent to taking the true gradient of a function which has been boxcar-smoothed along each parameter. This may more closely resemble existing measures of deviation than you like.

You should also consider the relationship to an evolutionary strategies style gradient estimate, which similarly provides an unbiased gradient estimate for a smoothed function, and which allows that estimate to be computed with fewer samples (at the cost of higher error).

Sec. 4.2 / Figure 3:
The results in this section will be *highly* sensitive to the choice of epsilon. You should discuss this, specify the epsilon used, and experimentally explore the dependence on epsilon.

"The results indicate that the cosine similarity between coarse gradient and CDG can explain the relationship between gradient mismatch and performance of model better than previous approaches. "
Don't know that I followed this. Gradient mismatch is never formally defined, so it's hard to know what this says about its relationship. Additionally, CDG sounds more like something which is correlated with, rather than an explanation for, performance.

" cosine similarity between coarse gradient and CDG can explain the relationship between gradient mismatch and performance of model better " --> " cosine similarity between coarse gradient and CDG can explain the relationship between gradient mismatch and performance of model better "

"we shift the bias of BN layer which comes right before the activation function layer. "
Did you try using these bias values without pre-training as a ternary network? I suspect it would work just as well!

"Please note that BN layers followed by binary activation layer can be merged to the threshold of the binary activation layer, incurring no overhead at inference stage."
Did not understand this.

"it is expected that the fine-tuning increases the accuracy even further"
Does it improve the accuracy further? Should state this as result, not prediction, and should have an ablation experiment showing this.

"Table 2 shows the validation accuracy of BNN in various schemes."
Why not test accuracy?

Figure 6:
What are the filled circles?
What was the sampling grid for the HP search? The images have high spatial frequency structure that I suspect is an artifact of the interpolation function, rather than in the data.

----

Update post-rebuttal:

The authors have addressed the majority of my concerns, through both text changes and significant additional experiments. I am therefore increasing my score. Thank you for your hard work!

**Experience Assessment:**

I have read many papers in this area.

**Review Assessment: Checking Correctness Of Derivations And Theory:**

I assessed the sensibility of the derivations and theory.

**Review Assessment: Checking Correctness Of Experiments:**

I assessed the sensibility of the experiments.

**Review Assessment: Thoroughness In Paper Reading:**

I read the paper at least twice and used my best judgement in assessing the paper.

---

> ### Author Response · Authors · 2019-11-12
> **Response to Reviewer #4 (part 4)**
>
>
> Q11: "it is expected that the fine-tuning increases the accuracy even further" Does it improve the accuracy further? Should state this as result, not prediction, and should have an ablation experiment showing this.
>
> A11: We agree to state this as result rather than prediction. Our experimental results (Figure 6 in Section 6.2) indeed show that the fine-tuning procedure actually increases the accuracy further. Please note that decoupling the ternary model without the fine-tuning does not change computation results of the network. Therefore, the accuracy of the decoupled binary model is same as that of the coupled ternary model when fine-tuning is not applied. For example, in case of VGG-7 on CIFAR-10 dataset, the accuracy of decoupled binary model without fine-tuning is 89.69% which is same as that of the coupled ternary model (shown in Section 6.1). In contrast, during the fine-tuning process, the weight for each of the decoupled binary network is tuned separately. The results for VGG7 on CIFAR-10 dataset shows that 0.75% of accuracy improvement can be achieved by fine-tuning which results in 90.44% accuracy (shown in Section 6.1 again).
>
> The improvement was also observed on ImageNet dataset. Accuracy results before and after fine-tuning for AlexNet, ResNet-18, and ResNet-18(+sc) are shown below.
> =================================================================
> |					|	Before	fine-tuning	|	  After fine-tuning	|
> |          Network		|    Top-1(%)	|    Top-5(%)	|    Top-1(%)	|    Top-5(%)	|
> ---------------------------------------------------------------------------------------------------------
> |           AlexNet		|        50.7	|        74.4	|        52.7	|        76.0	|
> |        ResNet-18		|        58.8	|        81.3	|        60.4	|        82.3	|
> |     ResNet-18(+sc)	|        59.1	|        81.3	|         60.9	|        82.6	|
> =================================================================
>
> To reflect the update, we revised the sentence as follows.
> “experimental results show that the fine-tuning increases the accuracy even further. Detailed experimental results will be discussed in the next section.”
>
>
>
> Q12: "Table 2 shows the validation accuracy of BNN in various schemes." Why not test accuracy?
>
> A12: Table 2 shows the validation accuracy of BNN in various schemes on the ImageNet Large Scale Visual Recognition Challenge (ILSVRC) 2012 dataset. ImageNet dataset has 1.28M images for training set and 50K images for validation set. Because the test set is not available to the public, (the test set is privately used for the ImageNet Large Scale Visual Recognition Challenge competition only), all the results of previous work in Table 2 are validation accuracy. For fair comparison with previous results, we also provided validation accuracy for our scheme.
> If required, it is possible to split the provided training set to train/valid sets and to use the provided validation set as a test set. However, the test accuracy from the experiment cannot be fairly compared with other results. Nevertheless, if reviewer asks for the test accuracy, we are willing to conduct addition training and report the test accuracy. Please let us know.
>
>
>
> Q13: Figure 6: What are the filled circles? What was the sampling grid for the HP search? The images have high spatial frequency structure that I suspect is an artifact of the interpolation function, rather than in the data.
>
> A13: We apologize for lack of detailed description of the figure. We mistakenly moved the information along with other details to the Appendix A while we made the original submission fit to 8-page limit.
> For the experiment on CIFAR-10 dataset, we tried 13 weight decay values (from 1e-6 to 1e-2) and 10 different initial learning rates (from 1e-4 to 1e-1). Therefore, 130 different data are plotted in each contour plot. Since our goal for hyper-parameter search is to ensure that we are not with completely wrong hyper-parameters, we believe that this amount of sampling grid is large enough for our experiment. The 5 circles represent top 5 test accuracy points and the red circle is for the best result.
>
> We updated the caption of Figure 6 as follows.
> “Figure 6: Training results of VGG-7 on CIFAR-10 dataset in the order of coupled ternary model, decoupled binary model and decoupled binary model trained from scratch (left). 5 circles represent the top 5 test accuracy points and the red circle is for the best result. The best accuracy result is shown at the top left corner of each contour plot. Test accuracy of various models with different activation precision and training schemes (right).”

---

> ### Author Response · Authors · 2019-11-12
> **Response to Reviewer #4 (part 3)**
>
>
> Q8: "The results indicate that the cosine similarity between coarse gradient and CDG can explain the relationship between gradient mismatch and performance of model better than previous approaches. " Don't know that I followed this. Gradient mismatch is never formally defined, so it's hard to know what this says about its relationship. Additionally, CDG sounds more like something which is correlated with, rather than an explanation for, performance.
>
> A8: We agree that CDG is correlated with performance rather than giving an explanation for performance. So we revised the above sentences as follows.
> "The results indicate that the cosine similarity between coarse gradient and CDG can provide better correlation with the performance of a model than previous approaches do. "
>
>
>
> Q9: "we shift the bias of BN layer which comes right before the activation function layer. " Did you try using these bias values without pre-training as a ternary network? I suspect it would work just as well!
>
> A9: Per reviewer’s suggestion, we conducted additional experiments in which, instead of using ternary pretrained model as initialization, we initialized the decoupled binary model with the shifted bias values used in section 5.1. For fair comparison, we tried the same amount of hyper-parameter search as that for the case shown in Figure 6. The results show that initializing the decoupled binary model with shifted bias still shows lower accuracy than using pre-trained coupled ternary model as initialization while it achieves a small increase in accuracy compared to initializing the model with zero bias values.
> Comparison among various training results of different schemes are shown below.
> =====================================================================
> Scheme												|     Best Accuracy	|
> ----------------------------------------------------------------------------------------------------------------
> Baseline binary										|		89.07%		|
> Coupled ternary										|		89.69%		|
> Decoupled binary with ternary initialization				|		90.44%		|
> Decoupled binary from scratch						|		88.93%		|
> Decoupled binary from scratch with shifted bias values	|		89.21%		|
> =====================================================================
>
>
> Q10: "Please note that BN layers followed by binary activation layer can be merged to the threshold of the binary activation layer, incurring no overhead at inference stage." Did not understand this.
>
> A10:  In BNN inference, computations for Batch-Normalization (BN) and the binary activation can be merged as a function which compares the weighted-sum value with a threshold [R1]. Therefore, modulating the BN bias with different values as in Eq. 5 does not incur additional overhead at inference stage.
> To avoid confusion, we revised the sentence as follows.
>
> “Please note that computations for BN  and the binary activation can be merged to a thresholding function. Therefore, calculating BNs with different bias values does not incur additional overhead at inference stage.”
>
> We described the merging process in detail for your information below.
>
> Let $X$ be a weighted-sum vector. Applying batch-normalization on $X$ results in $Y$ as in Eq. R1.
>
> $Y = \gamma*(X-\mu)/\sigma+\beta$            (Eq. R1)
>
> Here, $\gamma$,$\beta$,$\mu$, and $\sigma$ denote for weight, bias, mean, and standard deviation of the batch-normalization layer after training is finished. Then, the batch-normalization output $Y$ goes through binary activation function producing a binary output $Z$.
>
> $Z = +1 (\text{if}\;\;  Y \geq 0),  -1 (\text{else})$			  (Eq. R2)
>
> Here we formulate the binary activation function as the sign function for concise expression, but the same development is possible with different binary activation functions without loss of generality. At inference stage, Eq. R1 and Eq. R2 can be merged as Eq. R3  since batch-normalization parameters are fixed.
>
> $Z = +1 (\text{if} \;\;  X \geq \mu-\beta*\sigma/\gamma),  -1 (\text{else})$		(Eq. R3)
>
> In this way, batch-normalization can be merged to binary activation layer, incurring no overhead at inference stage.
> Furthermore, scaling factor for weights can also be merged to the binary activation function similar to the batch-normalization.
> This method has already been used in many previous works on BNNs [R1,R2].
>
> [R1] Umuroglu, Yaman, et al. "Finn: A framework for fast, scalable binarized neural network inference." Proceedings of the 2017 ACM/SIGDA International Symposium on Field-Programmable Gate Arrays. ACM, 2017.
> [R2] Liu, Zechun, et al. "Bi-real net: Enhancing the performance of 1-bit cnns with improved representational capability and advanced training algorithm." Proceedings of the European Conference on Computer Vision (ECCV). 2018.

---

> ### Author Response · Authors · 2019-11-12
> **Response to Reviewer #4 (part 2)**
>
>
> Q6: eq. 3: Note that this equation is equivalent to taking the true gradient of a function which has been boxcar-smoothed along each parameter. This may more closely resemble existing measures of deviation than you like. You should also consider the relationship to an evolutionary strategies style gradient estimate, which similarly provides an unbiased gradient estimate for a smoothed function, and which allows that estimate to be computed with fewer samples (at the cost of higher error).
>
> A6: Thank you very much for pointing out this important point. We agree that the CDG is equivalent to the gradient of the smoothed loss function. We think their equivalence can provide better explanation about the theoretical background of CDG. Therefore, we revised section 4.1 to introduce CDG with theoretical background based on the smoothed loss function. Please note that our intention was to provide a proper gradient measure as an alternative to the true gradient for gradient mismatch estimation.
> In that respect, we think that the basic philosophy of evolutionary strategy-style (ES) gradient estimator is very similar to that of our CDG (Eq. 3) since the ES gradient estimator also provides (estimated) gradient for a smoothed function as the reviewer correctly described. We were not aware of the ES gradient estimate when writing the manuscript, and after a quick survey of the literature for the ES gradient estimate, we believe that ES gradient estimator can be another good candidate for the quantitative assessment of the gradient mismatch.
>
> To verify, we measured the cosine similarity between the ES gradient estimator and the coarse gradient and observed that the results show a similar trend with the case using CDG (Figure 15 in Appendix H). Most notably, the ES gradient estimator-based assessment also shows that there is a large gap in the cosine similarity between the ternary activation case and the binary activation case with any STEs.
>
> Due to the time limitation in rebuttal period, we cannot thoroughly assess the relative strengths and weaknesses of the CDG estimator-based approach compared to the ES gradient estimator-based approach. However, we plan to continue to study the two approaches as the comparative assessment of the two approaches may open up an opportunity to find more sophisticated and practical gradient mismatch assessment methodology for quantized activation neural network.
>
> In this study, our main goal for the derivation of CDG was to have a solid justification to apply ternary activation neural network to train binary activation neural network, so the similar results from another approach further strengthens our main motivation for developing the BinaryDuo scheme.
>
> To reflect the update, we revised the manuscript as follows.
>
>     - First, we changed the title of section 4.1 to “Gradient of Smoothed Loss Function” from “Coordinate Discrete Gradient”. We also updated section 4.1 to introduce CDG with theoretical background based on the smoothed loss function.
>     - We replaced the term “CDG” with “gradient of smoothed loss function” in the abstract and conclusion.
>     - We updated Eq. 3 to explain the equivalence of  the gradient of smoothed loss function and CDG.
>     - At the end of section 4.1, we added the following sentence: “Note that the evolutionary strategy-style gradient estimation is another good candidate which can be used for the same purpose as it provides an unbiased gradient estimate for a smoothed function in a similar way (Choromanski et al., 2018). Please refer to Appendix H for more information.”
>     - At the end of section 4.2, we added the following sentence: “We also measured the cosine similarity between the estimated gradient using evolutionary strategies and the coarse gradient. The results show a similar trend to that of the results using CDG (Figure 15 in Appendix H). Most notably, the evolutionary strategy-based gradient mismatch assessment also shows that there is a large gap in the cosine similarity between the ternary activation case and the binary activation case with any STEs.”
>
> We thank the reviewer again to help us to generalize and strengthen the gradient estimate flow to provide the strong justification to develop the BinaryDuo scheme.
>
>
>
> Q7: Sec. 4.2 / Figure 3: The results in this section will be *highly* sensitive to the choice of epsilon. You should discuss this, specify the epsilon used, and experimentally explore the dependence on epsilon.
>
> A7: We measured the cosine similarities for various epsilon values (epsilon = 1e-4 to 1e-2) and the results show that overall trend is maintained regardless of the epsilon value although the absolute values change depending on the epsilon value.  Detailed experimental results with various epsilon values are given in Appendix C.1).

---

> ### Author Response · Authors · 2019-11-12
> **Response to Reviewer #4 (part 1)**
>
> Thank you very much for your constructive comments. We could improve the quality of our work significantly by responding to your comments.
>
>
> Q1: "Binary Neural Network (BNN) has been gaining interest thanks to its computing cost reduction and memory saving." --> "Binary Neural Networks (BNNs) have been garnering interest thanks to their compute cost reduction and memory savings." (will stop making English language corrections from here on)
>
> A1: Thanks very much for pointing out grammatical errors in our writing. We updated some sentences including the above one  in the revised manuscript. We will do our best to improve the writing in the final manuscript.
>
>
>
> Q2: "Therefore, we argue that the sharp accuracy drop for the binary activation stems from the inefficient training method, not the capacity of the model." This could also be due to poor initialization in the binary case. e.g., it might make sense to initialize the binary network with bias=-0.5, so that the nonlinearity has a kink at pre-activation=0, rather than pre-activation=0.5.
>
> A2:  Per reviewer’s suggestion, we conducted more experiment on CIFAR-10 dataset by initializing the binary network with bias=0.5 so that the nonlinearity has a kink at pre-activation=0. Please note that bias=0.5 instead of -0.5 needs to be used to make the nonlinearity have a kink at pre-activation=0 for the current activation function because each pre-activation needs to be shifted by 0.5 after the bias value is added.
> We trained four different networks with different width factors (x1, x1.25, x1.5 and x2) and each network was trained for 4 runs and the mean results are reported below.
> ---------------------------------------------------------------------------------------
> |		|					width factor					|
> |   Bias	|	x1		|	x1.25	|	x1.5		|	x2		|
> ---------------------------------------------------------------------------------------
> |     0	|	89.07	|	89.22	|	89.41	|	89.60	|
> |     0.5	|	88.96	|	89.32	|	89.58	|	89.69	|
> ---------------------------------------------------------------------------------------
> As shown in the table above, we observe that initializing with bias=0.5 does not significantly improve the results.
> In addition, the sharp accuracy drop for the binary activation was also observed in many previous works where symmetric signum function was used for binary activation function (e.g. ABC-Net as shown in Table 1 in our paper) for which nonlinearity has a kink at pre-activation=0. Therefore, we believe that using bias value of 0 is not the reason for sharp accuracy drop.
>
>
>
> Q3: "Unfortunately, it is not possible to measure the amount of gradient mismatch directly because the true gradient of a quantized activation function is zero almost everywhere. " It *is* possible to measure the mismatch to the true gradient exactly. One could even train using the true gradient. It's just that the true gradient is useless.
>
> A3: We agree that it is possible to calculate the true gradient. As the reviewer mentioned, the main point should have been that the measured results will not be “useful” because the value will be zero almost everywhere. We replaced the word “possible” with “useful” in the revised draft as follows.
> “Unfortunately, it is not useful to measure the amount of gradient mismatch directly because the true gradient of a quantized activation function is zero almost everywhere."
>
>
>
> Q4: Fig 1b -- this is a nice baseline.
>
> A4: Thanks very much for your encouraging comments.
>
>
>
> Q5: "the steepest descent direction, which is the direction toward the point with the smallest loss at given distance" This is not the usual definition of steepest descent direction. If you're going to redefine this, should do so mathematically and precisely (for instance, you are going to run into trouble with the word "distance", since your coordinate discrete gradient more closely resembles an L\infty-ball perturbation, rather than an L2-ball perturbation.
>
> A5: To get rid of the confusion over the terminology, we decided not to use the term “steepest descent direction”  and updated the sentence in the revised manuscript as follows.
>
> (original) “Since the true gradient of quantized activation network is zero almost everywhere, we cannot use the true gradient to find the steepest descent direction, which is the direction toward the point with the smallest loss at given distance.”
>
> (revised) “Since the true gradient of quantized activation network is zero almost everywhere, using the value of the true gradient does not provide a useful measure of the gradient mismatch problem.“

---

### Official Review · AnonReviewer2 · 2019-11-06
**Official Blind Review #2**

**Rating:** 6

**Review:**

To be honest, I only read several papers on the neural network quantization. I am not familiar with this research topic, so I  provide my judgement based on my own limited knowledge rather than thorough comparison with other related works.

1. The motivation is clear. The 1-bit activation networks usually deteriorates the performance greatly.
2. The gradient mismatch for discrete variable did bring difficult for optimization. Do you mean 1-bit activation has larger gradient mismatch than other bits, at least in the defined cosine similarity by this paper?
3. As to Eq(3), Appendix C.1 describes the way to choose step size. I understand the logic, but for the detailed method, is it cross-validation with grid search or some other tricks?
4. Is there any relation between the decoupling method in Section 5 and the proposed estimated gradient mismatch in Section 4.2?

**Experience Assessment:**

I do not know much about this area.

**Review Assessment: Checking Correctness Of Derivations And Theory:**

I assessed the sensibility of the derivations and theory.

**Review Assessment: Checking Correctness Of Experiments:**

I assessed the sensibility of the experiments.

**Review Assessment: Thoroughness In Paper Reading:**

I read the paper at least twice and used my best judgement in assessing the paper.

---

> ### Author Response · Authors · 2019-11-12
> **Response to Reviewer #2**
>
> Thank you very much for your constructive comments. We could improve the quality of our work significantly by responding to your comments. Below, we summarize your questions and address them in order.
>
>
> Q1: The motivation is clear. The 1-bit activation networks usually deteriorates the performance greatly.
>
> A1: Thank you very much for the positive comment.
>
>
>
> Q2: The gradient mismatch for discrete variable did bring difficult for optimization. Do you mean 1-bit activation has larger gradient mismatch than other bits, at least in the defined cosine similarity by this paper?
>
> A2: Yes. We think that 1-bit activation has larger gradient mismatch than other bit cases because the cosine similarity between the gradient estimate and the coarse gradient shows a large difference between ternary and 1-bit activation cases while similar cosine similarity values are exhibited for 1-bit activation cases with various STEs (Fig. 3)
>
>
>
> Q3: As to Eq. 3, Appendix C.1 describes the way to choose step size. I understand the logic, but for the detailed method, is it cross-validation with grid search or some other tricks?
>
> A3: We measured the cosine similarities for various epsilon values (epsilon = 1e-4 to 1e-2) and the results show that overall trend is maintained regardless of the epsilon value although the absolute values change depending on the epsilon value.  Detailed experimental results with various epsilon values have been added in Appendix C.1.
>
>
>
> Q4. Is there any relation between the decoupling method in Section 5 and the proposed estimated gradient mismatch in Section 4.2?
>
> A4: The results from gradient mismatch analysis confirm that there is a large accuracy gap between ternary activation neural network and binary activation neural network with any STEs. The results gave us strong justification to apply ternary activation neural network to train binary activation neural network, which is a key idea in the decoupling method used in the proposed BinaryDuo scheme.

---

### Author Response · Authors · 2019-10-08
**code submission**

As the ICLR policy strongly recommends the code submission, we prepared an anonymized link for our code. The code can be accessed in the following link: https://drive.google.com/open?id=1NxZdaSB7gZPMVH35hqp1xaqZ7ilwVAtD

---

> ### Author Response · Authors · 2020-01-10
> **Updated code link**
>
> We moved the source code to a github repository. Please find the code at the following link:
> https://github.com/Hyungjun-K1m/BinaryDuo

---

### Decision · Program_Chairs · 2019-12-19

**Decision:**

Accept (Poster)

**Comment:**

Three reviewers suggest acceptance. Reviewers were impressed by the thoroughness of the author response. Please take reviewer comments into account in the camera ready. Congratulations!